# Bottom—up approach for mitigating extreme events with limited intervention options: a case study with Lorenz 96 model

Takahito Mitsui<sup>1</sup>, Shunji Kotsuki<sup>2,3</sup>, Naoya Fujiwara<sup>4</sup>, Astushi Okazaki<sup>2,3</sup>, and Keita Tokuda<sup>1</sup>

**Correspondence:** Takahito Mitsui (takahito321@gmail.com)

Abstract. Prediction and mitigation of extreme weather events are important scientific and societal challenges. Recently, Miyoshi and Sun (2022) proposed a control simulation experiment framework that assesses the controllability of chaotic systems under observational uncertainty, and within this framework, Sun et al. (2023) developed a method to prevent extreme events in the Lorenz 96 model. However, since their method is primarily designed to apply control inputs to all grid variables, the success rate decreases to approximately 60% when applied to a single site, at least in a specific setting. Herein, we propose an approach that mitigates extreme events by updating local interventions based on multi-scenario ensemble forecasts. Our method achieves a high success rate, reaching 94% even when applying interventions at one site per step, albeit with a moderate increase in the intervention cost. Furthermore, the success rate increases to 99.4% for interventions at two sites. Unlike control-theoretic approaches adopting a top-down strategy, which determine inputs by optimizing cost functions, our bottom-up approach mitigates extreme events by effectively utilizing limited intervention options.

#### 1 Introduction

Global warming has likely increased the frequency and intensity of extreme weather events worldwide, such as heatwaves and heavy rainfall (IPCC, 2023). These threats are expected to escalate during the present century. Efforts to mitigate the risks posed by extreme events include advancements in weather forecasting and the development of disaster-resilient infrastructures. More ambitious projects have focused on active manipulation of weather systems. A prominent example is Project STORMFURY conducted by the U.S. government from 1962 to 1983, which aimed to weaken tropical cyclones by seeding them with silver iodide, although the initially promising results of this study were questioned in later studies (Willoughby et al., 1985).

Artificial weather control faces several fundamental challenges. First, meso- or synoptic-scale weather systems, such as stationary rainbands and tropical cyclones, are significantly larger in scale than what we can feasibly influence through interventions. Second, weather systems are inherently chaotic and sensitive to the initial conditions, limiting system predictability. Observational uncertainties and discrepancies between models and reality further increase the complexity (Palmer and Hagedorn, 2006). Consequently, predicting the outcomes of interventions is inherently difficult. This paper proposes a mathematical approach to alleviate these challenges.

<sup>&</sup>lt;sup>1</sup>Faculty of Health Data Science, Juntendo Univerity, Urayasu, Japan

<sup>&</sup>lt;sup>2</sup>Institute for Advanced Academic Research, Chiba University, Chiba, Japan

<sup>&</sup>lt;sup>3</sup>Center for Environmental Remote Sensing, Chiba University, Chiba, Japan

<sup>&</sup>lt;sup>4</sup>Graduate School of Information Sciences, Tohoku University, Sendai, Japan

Several mathematical algorithms have been proposed for mitigating extreme weather events through interventions. Henderson et al. (2005) proposed a method using the four-dimensional variational data assimilation to identify the smallest temperature increments required for minimizing wind damage during a hurricane. Miyoshi and Sun (2022) proposed a framework called the Control Simulation Experiment (CSE), which extends the concept of observing system simulation experiments to investigate the controllability of dynamical systems under observational uncertainty. Their CSE successfully prevents regime transitions in the Lorenz 63 model, whereas the CSE application of Sun et al. (2023) reduces extreme events in the Lorenz 96 model. More recently, Kawasaki and Kotsuki (2024) introduced the model predictive control (MPC) within the CSE and applied it to controlling the Lorenz 63 model (see also a paper by Nagai et al. (2024)). However, control methods such as the MPC are computationally expensive. The computational cost of determining the optimal inputs can bottleneck practical applications of high-dimensional dynamical systems such as numerical weather predictions models. To overcome this bottleneck, Sawada (2024a, b) introduced an ensemble Kalman filter–based control method, which efficiently finds local and intermittent interventions, and applies it to controlling the Lorenz 63 and 96 models.

Following these previous works, we explore a mathematical algorithm for mitigating extreme events, using the Lorenz 96 model as a testbed. Practical weather control applications must be feasibly implementable. Given the large scales of weather systems and the limited energy available for control, interventions will be typically constrained to local sites and specific time-points (Sawada, 2024b). The Lorenz 96 model is a minimal mathematical system for investigating feasible control strategies in spatially extended chaotic systems (Lorenz, 1996; Sun et al., 2023; Sawada, 2024b). Sun et al. (2023) proposed a method for mitigating extreme events in the Lorenz 96 model. Their method is primarily designed to apply control inputs across all grid variables. When control is limited to a single site, the success rate of their method declines to approximately 60%, at least in a specific setting. To improve the results of Sun et al. (2023), we propose a new methodology that identifies effective local interventions.

Control-theoretic methods such as MPC are top-down approaches that compute optimal control inputs by minimizing a cost function subject to given constraints (Kawasaki and Kotsuki, 2024; Nagai et al., 2024; Ohtsuka et al., 2025). In contrast, the present study pursues a bottom-up approach to mitigate extreme events using a finite set of available intervention options. We assume that available options for human intervention in weather will remain limited for the foreseeable future, given the time required for technological advancements. Hence, we generate ensemble forecasts for a limited set of local intervention scenarios and implement the most effective intervention based on these forecasts. To account for the chaotic uncertainty in predictions, the selection of the "best" scenario is updated over time. The large number of potential intervention scenarios is reduced to a manageable subset to ensure computational feasibility. Despite this constraint, our method mitigates extreme events in the Lorenz 96 model with high success rates.

45

55

The remainder of this article is organized as follows. The Lorenz 96 model, the local ensemble transform Kalman filter (LETKF)-based data assimilation method, and our control algorithm for extreme events are explained in Sections 2.1–2.3, respectively. Section 3 presents the results, and Section 4 discusses the limitations and potential extensions of our method. We also highlight the similarity between our method and the so-called rare event algorithms, which efficiently simulate events with extremely low probabilities (Ragone et al., 2018; Wouters et al., 2023; Cini et al., 2024; Sauer et al., 2024).

Figure 1. Extreme events in the Lorenz 96 model: (a) spatiotemporal variations of  $X_j(t)$  over 365 d. The two arrows indicate extreme events exceeding the threshold value 14.217. (b) Time series of the maximum value across 40 sites at each time step. The horizontal red line represents the threshold 14.217. (c) Histogram of 6-h maxima, representing the maximum X-values across all the sites within each 6-h period.

## 2 Methods

60

#### 2.1 Lorenz 96 model and extreme events

The toy model of weather systems proposed by Lorenz (1996) contains J variables,  $X_1, ..., X_J$  and is governed by

$$\frac{dX_j}{dt} = (X_{j+1} - X_{j-2})X_{j-1} - X_j + F,$$

where F is a constant parameter. Periodic boundary conditions  $X_{J-1} = X_{-1}$ ,  $X_J = X_0$ ,  $X_{J+1} = X_1$  are assumed. The variables  $X_j$  can be interpreted as unspecified meteorological quantities measured along a circle of constant latitude of the Earth. For common parameter values J = 40 and F = 8 (Lorenz and Emanuel, 1998), the dynamics are chaotic (Fig. 1a). The time unit is commonly assumed as 5 days (d) (Lorenz and Emanuel, 1998). The error doubling time of the Lorenz 96 model is then comparable to those of modern General Circulation Models. The Lorenz 96 equation is integrated using the forth-order Runge-Kutta method with a time step of  $\Delta t = 0.01$  corresponding to 1.2 hours (h).

The Lorenz 96 model exhibits sporadic "extreme events" characterized by very high values. The method of Sun et al. (2023) aims to prevent values of the variable  $X_j$  above a threshold of 14.217. On average, the maximum values across all the sites during each 6-h period (hereafter referred to as the 6-h maxima) exceed the threshold twice per year (Figs. 1a and b). Herein, we adopt the same threshold (14.217) to enable a straightforward comparison between our results and theirs. The histogram of 6-h maxima is shown in Fig. 1c.

### 2.2 Data assimilation method

- Data assimilation combines model simulations and noisy observations to estimate the state of a dynamical system. Data assimilation is essential in weather forecasting and was thus employed in previous weather control studies. As in Sun et al. (2023), we adopt the Local Ensemble Transform Kalman Filter (LETKF) (Hunt et al., 2007), a variant of the Ensemble Kalman Filter, for data assimilation due to its efficiency and scalability in high-dimensional settings such as numerical weather prediction models (Lien et al., 2017).
- Herein, we briefly outline the LETKF (Hunt et al., 2007; Kotsuki and Bishop, 2022). Consider a dynamical systems model providing ensemble forecasts  $\{x^{b(i)}\}_{i=1}^m$ , where m is the number of ensemble members and the suffix b stands for background or forecast. The LETKF transforms the ensemble forecasts  $\{x^{b(i)}\}_{i=1}^m$  into an analysis ensemble  $\{x^{a(i)}\}_{i=1}^m$ , whose mean  $\overline{x^a}$  minimizes the following cost function:

$$J(\boldsymbol{x}) = (\boldsymbol{x} - \overline{\boldsymbol{x}^b})^T (\boldsymbol{P}^b)^{-1} (\boldsymbol{x} - \overline{\boldsymbol{x}^b}) + (\boldsymbol{y} - H(\boldsymbol{x}))^T \boldsymbol{R}^{-1} (\boldsymbol{y} - H(\boldsymbol{x})),$$

- where  $\overline{x^b}$  is the forecast mean,  $P^b$  is the forecast-error covariance matrix, y and H denote the observation vector and observation operator, respectively, and R is the observational-error covariance matrix (here assumed as diagonal). The first term in J(x) penalizes deviations of the analysis mean from the forecast, weighted by the confidence in the forecast. The second term penalizes deviations from the observations, weighted by the confidence in the observations.
- The LETKF optimizes the cost function J(x) through an ensemble-based approach and localization principles. The background error covariance is approximated as  $P^b \approx \frac{1}{m-1} X^b (X^b)^T$ , where  $X^b$  is the matrix of background ensemble perturbations with the i-th column given by  $x^{b(i)} \overline{x^b}$ . Equivalently, this expression can be written as  $P^b \approx Z^b (Z^b)^T$ , where  $Z^b = X^b / \sqrt{m-1}$ . Based on observations, the analysis ensemble is updated in the reduced-dimensional subspace spanned by  $Z^b$ . In particular, the LETKF updates the analysis on a grid by grid basis as

$$m{X}^a = \overline{m{x}^b} \cdot \mathbf{1} + m{Z}^b m{T},$$

$$T = \tilde{\boldsymbol{P}}^a (\boldsymbol{Y}^b)^T \boldsymbol{R}_{\text{loc}}^{-1} (\boldsymbol{y} - H(\overline{\boldsymbol{x}^b})) \cdot \boldsymbol{1} + \sqrt{m-1} (\tilde{\boldsymbol{P}}^a)^{1/2}.$$

Here, 1 is a row vector of ones, and  $\mathbf{Y}^b \approx (H(\mathbf{X}^b) - \overline{H(\mathbf{X}^b)} \cdot \mathbf{1})/\sqrt{m-1}$ . The matrix  $\tilde{\mathbf{P}}^a$  is derived as  $\tilde{\mathbf{P}}^a = \mathbf{\Lambda} \mathbf{D}^{-1} \mathbf{\Lambda}^T$  from eigenvalue decomposition  $(\tilde{\mathbf{P}}^a)^{-1} = \mathbf{I} + (\mathbf{Y}^b)^T \mathbf{R}_{\text{loc}}^{-1} \mathbf{Y}^b = \mathbf{\Lambda} \mathbf{D} \mathbf{\Lambda}^T$ . Further, the square root of  $\tilde{\mathbf{P}}^a$  is given as  $(\tilde{\mathbf{P}}^a)^{1/2} = \mathbf{\Lambda} \mathbf{D}^{-1/2} \mathbf{\Lambda}^T$ . The matrix  $\mathbf{R}_{\text{loc}}$  is the localized observation-error covariance, which is also diagonal. When updating at each grid point j, the i-th diagonal element of  $\mathbf{R}_{\text{loc}}^{-1}$  is defined as  $(\mathbf{R}^{-1})_{ii} \exp\left(-\frac{|i-j|^2}{2\sigma^2}\right)$  if  $|i-j| < 2\sqrt{\frac{10}{3}}\sigma$  and zero otherwise. Here,  $\sigma$  is the localization parameter. Localizing updates to relevant regions reduces the impact of spurious correlations in covariance estimates. At site j, only the j-th row of the analysis ensemble  $\mathbf{X}^a$  is used in subsequent forecasts. Error covariance underestimation is handled by a multiplicative covariance inflation as  $\mathbf{X}^b \leftarrow \rho \mathbf{X}^b$ , where  $\rho > 1$  is the inflation coefficient (Kotsuki and Bishop, 2022). In sum, the LETKF performs data assimilation locally and operates within a low-dimensional ensemble space, thereby avoiding expensive full-space computations and reducing the computational cost in high-dimensional systems. Although we use the LETKF in this study, comparable control performance may be achieved with other data assimilation

methods, such as the Serial Ensemble Square Root Filter, provided that state estimation is performed as accurately as with the LETKF (Miyoshi, 2005).

Following Sun et al. (2023), we observe the system at 6-h intervals (i.e., at every five time steps with  $\Delta t = 0.01$ ). The observation is modeled as  $y_j(t) = H(X_j(t)) + \varepsilon_j(t)$ , where  $\varepsilon_j(t)$  is independent white Gaussian noise with zero mean and unit variance. The linear observation operator  $H(X_j) = X_j$  is assumed at all 40 sites unless otherwise stated (partial observations are considered in Section 3.4). Every 6 h, having obtained the observations y, we assimilate the data using the LETKF to obtain the analysis ensemble  $\{x^{a(i)}\}_{i=1}^m$ , providing the initial condition of subsequent ensemble forecasts (cf. Fig. 2b(i)). Unless otherwise stated, we set m=10,  $\sigma=6.0$ ,  $\rho=1.03$ , and R as the identity matrix. Applying LETKF with these parameters, the root mean square error between the analysis mean and the true state is 0.1973, comparable to that of Sun et al. (2023) (0.1989).

## 115 2.3 Intervention method for reducing extreme events





To mitigate extreme events in the Lorenz 96 model, we introduce an intervention input  $u(t) = (u_1(t), u_2(t), ..., u_N(t))^T$  into the system as

$$\frac{dX_j}{dt} = (X_{j+1} - X_{j-2})X_{j-1} - X_j + F - u_j(t). \tag{1}$$

The sign preceding  $u_j(t)$  is negative by default. To identify effective intervention inputs, we perform ensemble forecasts assuming several sequences of intervention inputs u(t) called *intervention scenarios* (Fig. 2a). The *intervention-off scenario* planned during  $s \le t 

Figure 2. Method for reducing extreme events. (a) Intervention-off scenario and intervention scenarios at 40 possible intervention sites. Each cross mark (" $\times$ ") indicates the addition of a nonzero input at one of the 40 sites at each time step. In the intervention-off scenario (a-i), any intervention is turned off after 6 h. In an intervention scenario (a-ii), a new intervention site is selected after 6 h. The lower panels in both (a-i) and (a-2-ii) show the case where no intervention was selected in the previous 6-h cycle. (b) Key steps of the method: (b-i) data assimilation using the LETKF, (b-ii) risk assessment of a T-d ensemble forecast under the intervention-off scenario, and (b-iii) multi-scenario ensemble forecasts and interventions. Under the best scenario (blue), the worst member in the T-d ensemble forecast (thick arrow) is most effectively mitigated. See text for details.

Risk assessment. To avoid unnecessary interventions in the system, we preassess the risk of extreme events using a T-d ensemble forecast under the intervention-off scenario, based on the latest analysis ensemble (Fig. 2b(ii)). Focusing on a prediction horizon of T=7 d, we examine the dependence of the success rate on T. A binary flag "ALERT" signals the need for control operations. If ALERT = OFF and an extreme event exceeding the upper threshold 14.217 is predicted within T d in any ensemble member, we set ALERT = ON. If ALERT = ON and the maximum value across all the ensemble members within T d is below the lower threshold 13.5, we set ALERT = OFF. Otherwise the flag is unchanged.



Multi-scenario ensemble forecasts and local intervention. The action taken depends on the risk assessment result. If ALERT = OFF, we operate the intervention-off scenario given by Eqs. (1) and (2) in both the nature run and the ensemble forecast in the data assimilation using the LETKF. If ALERT = ON, we perform T-d ensemble forecasts under possible intervention scenarios (Fig. 2b(iii)). These multi-scenario ensemble forecasts are conducted less frequently (at 24-h intervals) than the 6-h

data assimilation because multi-scenario ensemble forecasts are computationally expensive and the forecast results change relatively infrequently. When planning a one-site intervention among the 40 sites, 400 forecasts with different combinations of 10 ensemble members and 40 intervention scenarios (i.e., sites) are performed. In each scenario, we obtain 10 outcomes corresponding to the 10 ensemble members. From the perspective of risk hedging, we assume that the best intervention scenario minimizes the maximum value across all the sites and ensemble members over the next T d. Other criteria, such as minimizing the ensemble-mean of the maximum value across all sites, are also possible. If any intervention scenario with nonzero u cannot reduce the maximum from that of the intervention-off scenario, the intervention-off scenario becomes the best scenario. The selected best scenario is then operated from the next 6-h cycle.

# 3 Results



This section presents the results of CSE using our method. A single CSE spans 1010 years (yr). Transients are eliminated by excluding the first 10 y, and analysis is performed by dividing the subsequent 1000 yr (730,000 steps) into ten 100-yr segments. The performance scores are assessed in each segment, and the standard deviations of the scores are represented as error bars in the corresponding figures (we have verified that similar error bars are obtained using an alternative sampling approach in which one hundred 100-yr samples are extracted from the 1000-yr series using a 10-yr sliding window). As the error doubling time is 0.42 units, i.e., 42 steps, the segments are effectively independent. Following Sun et al. (2023), the simulation run controlled by intervention inputs is called the *controlled nature run*.

Our control strategies are evaluated in terms of three metrics. The first metric is the *success rate*, defined as follows (Sun et al., 2023):

success rate = 
$$1 - \frac{\text{(number of 6-h intervals in 100 yr with extreme events)}}{200}$$

Here, the denominator is set to 200 because two extreme events per year are expected at a threshold of 14.217. Alternatively, the success rate can be defined by the ratio of extreme event frequencies with and without control, but both definitions yield nearly identical results. The second metric is the *intervention energy* defined by the sum of displacement norms induced by intervention inputs (Sun et al., 2023). It is mathematically written as  $\sum_t \|X_{\text{intervened}}(t + \Delta t) - X_{\text{unintervened}}(t + \Delta t)\|$ , where  $X_{\text{intervened}}(t + \Delta t)$  and  $X_{\text{unintervened}}(t + \Delta t)$  are one-step-ahead values for the cases with and without intervention inputs, respectively, both computed from the controlled nature run  $X_{\text{intervention}}(t)$ . For one-site intervention, this metric is roughly estimated as

intervention energy  $\approx u\Delta t \times$  (number of steps with nonzero intervention).

The third metric is the *number of scenario changes*. It measures the frequency of changes in the control input vector, u(t), per year in the controlled nature run, serving as a measure of operational cost. This metric is particularly relevant when implementing interventions by obstacles requiring no energy postplacement.

Obviously, other types of cost functions can be considered—for example, changing the intervention site to a more distant location may incur higher costs. However, in this study, we focus on the three performance metrics mentioned above.

# 3.1 One-site intervention among 40 sites







First, we consider the simplest intervention case at a single site among the 40 sites at each time step. Figure 3 presents an example snapshot of the CSE with a prediction horizon of T=7 d. An extreme event exceeding the 14.217 threshold is detected in the 7-d ensemble forecast conducted at t=123, triggering ALERT = ON (Fig. 3a). Next, 7-d ensemble forecasts are performed for 40 scenarios, each targeting a different site (Fig. 3b). At t=123, the intervention at site 35 most effectively minimized the maximum across the sites and ensemble members over the ensuing 7 d. Therefore, the intervention at site 35 is implemented from the next 6-h cycle (Fig. 3d). The optimal intervention site is updated at 24 h intervals. The intervention is stopped once all 7-d ensemble forecasts fall below the lower threshold of 13.5. As a result, the 6-h maxima of the controlled nature run (blue) were successfully maintained below the upper threshold of 14.217 (Fig. 3e).

Figure 4 compares the histograms of 6-h maxima in the controlled and uncontrolled nature run for an input size of u=1.6 and a prediction horizon of T=7 d. The extreme events are mitigated at a success rate of 94%, largely exceeding the  $\sim$ 60% success rate achieved by Sun et al. (2023) at a specific setting of 4 d for the prediction horizon and 0.7 for the perturbation-size coefficient  $\alpha$ . Maintaining T=7 d, the success rate initially increases with increasing input size u and eventually saturates at around 94% when u exceeds  $\sim$  1.6 (Fig. 5a). However, increasing the input size u increases the required intervention energy because a larger force induces a greater displacement of the state, highlighting a success—cost trade-off. At  $u \gtrsim 0.5$ , our method requires higher intervention energy than Sun et al. (2023), though it remains below three times the energy required by their method (Fig. 5b). With the same intervention energy (in the case u=0.5), our success rate ( $\sim$ 45%) is lower than 60% reported in Sun et al. (2023). Fortunately, the number of scenario changes decreases with increasing input size u, reflecting the corresponding magnitude increase of the interventions (Fig. 5c). The frequency of scenario changes is reasonably small (20–27 per year).

The intervention size u=1.6 that achieves successful control (Fig. 4) is not particularly small relative to the parameter F=8 or the typical state range  $-12 \lesssim X_i \lesssim 16$  (Fig. 1a). This corresponds to an intervention-induced displacement at one step, with a size of  $u\Delta t=0.016$ , where  $\Delta t=0.01$ . In comparison, Sun et al. (2023) employed intervention-induced displacements of size  $\alpha D_0$ , ranging from 0.01989 to 0.1989. Thus, the intervention magnitude in our study is comparable to or smaller than that in Sun et al. (2023).

Next, the performance dependence on the prediction horizon T was examined for a fixed input size u. Increasing the prediction horizon T generally increases the success rate (Fig. 6a) but disadvantageably increases both the intervention energy and number of scenario changes (Figs. 6b and c), again highlighting the success—cost trade-off.

# 3.2 One-site intervention near the predicted extreme

Assuming real-world applications, interventions across all sites could be neither feasible nor necessary. Therefore, we restrict interventions to the vicinity of the site of a predicted extreme event under the intervention-off scenario. Figure 7 plots the three control-performance measures for different numbers of neighboring sites where interventions are allowed  $\{1, 3, 7, 11, 21, 31, 40\}$ . Interestingly, the success rate is maximized when the neighborhood size is 21, approximately half

Figure 3. A snapshot of the CSE with input size u=1 and prediction horizon T=7 d: (a) 7-d forecast from day 123 under the intervention-off scenario. The 10 lines represent the time evolution of the maximum values across the 40 sites,  $\max_{1 \le j \le 40} \tilde{X}_j(t)$ , for each of the 10 ensemble members. As one ensemble member crosses the threshold 14.217, ALERT is set to ON. (b) Same as (a) but for 40 different intervention scenarios. The best scenario is the intervention at site 35 (blue). (c) Predicted 6-h maximum over the ensuing 7 d. Each point is plotted at the time of forecasting. (d) Sites of predicted extremes (red circles) and interventions (crosses) during each 6-h cycle. Note that the actual intervention is implemented 6 h after the forecast, and the optimal intervention site is updated at 24 h intervals. (e) Controlled nature run, successfully maintained below the threshold of 14.217.

the total number of sites. From this result, we inferred that restricting the number of intervention sites prevents ineffective interventions at sites far from the extreme event.

Figure 4. CSE result of intervening at a single site among 40 sites. Plotted are histograms of the 6-h maxima during the controlled (blue) and uncontrolled (white) nature runs. The input size u is 1.6. The vertical red line represents the 14.217 threshold below which the state variables are intended to be maintained. The success rate is 94.2%.

Figure 5. Performance metrics versus input size u of the proposed control method at a single intervention site: (a) success rate, (b) intervention energy, and (c) number of scenario changes. The prediction horizon T is 7 d. Points represent the average values, and error bars indicate one standard deviation across 10 experimental results with different initial conditions. The horizontal dashed lines show the reference scores obtained under a specific setting in the work by Sun et al. (2023).

# 3.3 Two-site intervention in the vicinity of a predicted extreme

The previous subsections were limited to single-site interventions. This subsection analyzes the outcomes of intervention at two sites. Figure 8 shows the CSE results in a case with at most two intervention sites among 11 self-inclusive neighbors, where u = 2 and T = 7 d. The total number of possible scenarios is 67, comprising  $\binom{11}{2}$  combinations of two intervention sites, 11

Figure 6. Performance metrics versus prediction horizon T of the proposed control method at a single intervention site: (a) success rate, (b) intervention energy, and (c) number of scenario changes. Results are shown for different input sizes u = 1, 1.6, and 2. The horizontal dashed lines are the reference scores obtained under a specific setting in the work by Sun et al. (2023).

Figure 7. Performance of the proposed control method versus number of intervention-eligible sites at a single intervention site: (a) Success rate, (b) intervention energy, and (c) number of scenario changes. u = 2 and T = 7 d.

single-site interventions, and the no-intervention scenario. The success rate reaches 99.4%. Surprisingly, only a slight increase in the input dimensions efficiently improves the control performance.

Next, we examine the sensitivity of the results to number of intervention-eligible sites (Fig. 9). The success rate remains at 98%–99% when the number of intervention-eligible sites is five or higher and remains high (92.7%) even when the eligible-site number declines to three. Meanwhile, the interaction energy of multisite intervention increases by only around 15%–25% from that of single-site interaction (compare Fig. 9 with Fig. 7).

Figure 8. CSE result of intervening at two sites near a predicted weather extreme. Plotted are histograms of 6-h maxima during the controlled natural (blue) and uncontrolled (white) nature runs with u = 2.0 and T = 7 d. The vertical red line represents the 14.217 threshold below which the state variables are intended to be maintained. The success rate is 99.4%.

Figure 9. Performance metrics of the proposed control method with two intervention sites versus number of intervention-eligible sites: (a) success rate, (b) intervention energy, and (c) number of scenario changes. u = 2 and T = 7 d.

### 3.4 Partial observation


In real-world applications, complete observations of all state variables are often lacking. Therefore, this section investigates the performance of our method under partial observations. Following Sun et al. (2023) and Sawada (2024b), we assume that the state of the system can be observed at 20 sites with odd indices.

We first consider eligible intervention at a single site among 40 sites. Figure 10 compares the performance results of the partial and complete observations as functions of input size u. Partial observations decrease the success rate by 26% at most over  $0 \le u \le 2$ , while increasing the intervention energy and number of scenario changes by 57% and 68% at most, respectively.

Figure 10. Comparison of performance metrics versus input size u in cases of full and partial observations during single-site intervention among 40 sites: (a) success rate, (b) intervention energy, and (c) number of scenario changes. T = 7 d.

Second, we compare the performance results of partial and full observations when intervening at two sites among the 40 sites. Figure 11 plots the results as functions of number of intervention-eligible sites at u=2 and T=7 d. Partial observations decrease the success rate by a small fraction (5%–10%) from that of complete observations (Fig. 11a), but increase the intervention energy and number of annual scenario changes (Figs. 11b and c). Therefore, the present method handles partial observations with a high success rate while it incurs higher control costs. The success rate can be further improved by slight retuning of the LETKF parameters. For the localization length scale of  $\sigma=5.9$  and the number of ensemble members m=11, the reduction of the success rate due to partial observation becomes minimal (only 0%–4%) (Fig. 11a, triangle), while the re-tuning has little effect on intervention energy and number of annual scenario changes (Figs. 11b and c, triangle).

#### 4 Summary and discussions




Herein, we proposed a bottom-up approach that reduces extreme events in the Lorenz 96 model with limited intervention options. Alongside data assimilation, we performed T-d ensemble forecasts under the intervention-off scenario. When the T-d ensemble forecasts included an extreme event, we explored the best scenario that maximally mitigated the highest extreme value produced by the ensemble members through multi-scenario ensemble forecasts. We then implemented the identified best scenario. The success rate of our method was approximately 94% when perturbing one site per step, and 99.4% when perturbing two sites per step. Under the one-site intervention setting, our method improved the success rate of Sun et al. (2023) ( $\sim$ 60%) by approximately 34%, although it required several times more intervention energy. Several factors may account for this significant improvement. In Sun et al. (2023), the perturbation is scaled between the worst and best trajectories, implying that the control input is not optimized. In contrast, our method selects the optimal intervention scenario from available options. It should be mentioned that the reported success rate of approximately 60% in Sun et al. (2023) may not be fully optimized

Figure 11. Comparison of performance metrics versus the number of intervention-eligible sites under full and partial observations during two-site interventions: (a) success rate, (b) intervention energy, and (c) number of scenario changes. Here, u=2 and T=7 d. For the partial observation case, we show results using the default LETKF parameters with localization length scale  $\sigma=6.0$  and ensemble size m=10 (diamonds, solid line), as well as results using re-tuned parameters  $\sigma=5.9$  and m=11 (triangles, dashed line). In the latter case, the parameters are re-tuned to maximize the success rate.

with respect to the chosen parameters. Moreover, with the same intervention energy (e.g., for u = 0.5), our method yields a lower success rate of approximately 45%.





In addition to its high success rate, our method also exhibited robustness to missing observations, achieving high success rates even when half of the observations were unavailable (Fig. 11). Sensitivity analyses varying the method parameters revealed the success—cost trade-off. In the actual implementation, the parameters should be chosen to maximize the success rate while adhering to constraints on intervention energy, the number of scenario changes per year, and other relevant factors. Interestingly, both the maximization of the success rate and the minimization of intervention energy were achieved at an intermediate number of intervention-eligible sites (Fig. 7a). Limiting these sites not only helps avoid ineffective interventions but also reduces the computational cost of optimizing the intervention scenario.

The success rate of the method rather smoothly changes with the intervention size u (Figs. 5 and 10). However, when mitigating each extreme event, the success of mitigation can strongly depend on the input size u. To demonstrate the sensitivity to u, Fig. 12 plots the extreme value observed over a certain time interval as a function of u in an extreme-event instance. The maximum value drastically changes between u=1.21 and u=1.22 because the best intervention sites differ between the two cases, and this difference leads to different trajectory evolutions. Therefore, even if controlling an extreme event is predicted to fail at a particular value of the input size u, successful control may be achieved with a slight additional increase of u. This complex behavior arises from the interplay between algorithmic thresholds and nonlinear trajectory evolutions.

Our bottom-up approach may be particularly well-suited when intervention options can be limited or discretized. In contrast, top-down methods such as MPC are generally more effective when intervention options are continuous or nearly unlimited.

Figure 12. Sensitivity of the present control method to size of the control input u. Plotted is the 6-h maximum of the controlled nature run over a certain time horizon 218 < t < 228 as a function of u. The extreme value largely depends on u, especially between u = 1.21 and u = 1.22. The dashed line represents the 14.217 threshold.

Moreover, our method is relatively simple to implement and highly interpretable, whereas methods like MPC, although typically more computationally demanding, can provide mathematically optimal control solutions under given constraints.

In this work, we identified the best scenario among multiple candidates through a brute-force grid search, which limits the number of evaluable scenarios. To overcome this limitation, optimization techniques such as Bayesian optimization, genetic algorithms, and particle swarm optimization, which can efficiently identify effective intervention scenarios, will be explored in future work. Moreover, recent artificial-intelligence-based weather prediction models may enlarge the number of potential intervention scenarios (Price et al., 2025; Kotsuki et al., 2024).




Our control method is analogous to rare event simulation (RES), which simulates rare events with probabilities too small to be sampled through standard Monte Carlo simulations (Ragone et al., 2018; Wouters et al., 2023; Cini et al., 2024; Sauer et al., 2024). A type of RES called the genealogical particle analysis (Del Moral and Garnier, 2005) generates multiple system trajectories with small, naturally occurring perturbations (mutations) that preserve the statistical properties of the system. A subset of trajectories that approach the target event is selected and replicated with appropriate weights while all others are terminated (selection). By iterating the mutation and selection processes, this RES efficiently simulates rare events and estimates their true probabilities based on the number of simulated rare events adjusted by the assigned weights. The mutation and selection processes in RES are similar to our multi-scenario ensemble forecasts and the selection of intervention scenarios, respectively. In fact, our method was inspired by the RES framework, which obtains a rare event through a sequence of small perturbations. Analogously, our bottom—up approach mitigates extreme events through a sequence of limited intervention options.

At present, our work is a proof of concepts demonstrated on a toy model of weather systems. The method should be tested in higher-dimensional models such as realistic numerical weather models and in situations with model uncertainty.

Code availability. The codes used in this study will be uploaded to a repository following acceptance of the paper.

# Appendix A: Step-by-step description of the control simulation experiment

Denote by  $M(s) = \max_{1 \le j \le N, \ s \le t < s + T \ d, \ 1 \le i \le m} \tilde{X}_j^{(i)}(t)$  the predicted maximum value across all the sites and ensemble mem-290 bers over the next T d from time s. The following operations are performed at 6-h intervals.

- 1. At time s, we have the analysis ensemble  $\{x^{a(i)}(s)|i=1,...,m\}$  with m members (e.g., m=10). Starting from  $\{x^{a(i)}(s)|i=1,...,m\}$  with the input u(t) planned in the previous 6-h interval, we integrate Eq. (1) to obtain the background ensemble  $\{x^{b(i)}(s+6h)|i=1,...,m\}$ .
- 2. Starting from the true state x(s), compute the controlled nature run under the same input u(t). We obtain x(s+6h) and the 6-h maximum defined by  $\max_{1 \le i \le N, s \le t \le s+6h} X_i(t)$ .
  - 3. Generate noisy observations  $y_j(s+6h) = X_j(s+6h) + \varepsilon_j$ , where  $\varepsilon_j$  denotes independent white Gaussian noise with zero mean and unit variance.
  - 4. Assimilate the data using the LETKF described in Section 2.2, obtaining the analysis ensemble  $\{x^{a(i)}(s+6h)|i=1,...,m\}$  at s+6h.
- 5. Perform a T-d ensemble forecast under the intervention-off scenario (Section 2.3). Starting from  $\{x^{a(i)}(s)|i=1,...,m\}$  with the input in Eq. (2), integrate Eq. (1) to yield a T-d forecast  $\tilde{X}(t)$  over  $s \leq t < s + T$  d.
  - 6. If ALERT flag is OFF and one of the ensemble members in the T-d forecast exceeds the upper threshold 14.217 (that is, M(s) > 14.217), set ALERT=ON and ELAPSE=0 and go to step 9. Otherwise, proceed to step 7.
- 7. Else if ALERT flag is ON and all ensemble members in the T-d forecast remain below the lower threshold 13.5 (that is, M(s) < 13.5), set ALERT=OFF, choose the intervention-off scenario from the next 6-h cycles (t > s + 6 h), and return to step 1, advancing the time s by 6 h. Otherwise, go to step 8.
  - 8. Else if ALERT=ON, increment the elapsed time counter (ELAPSE) by 1 and go to step 9.
  - 9. If ALERT=ON and ELAPSE is zero or a multiple of 4 (that is, every 24 h), perform T-d ensemble forecasts of all intervention scenarios in Eq. (3). For each scenario, obtain the maximum M(s). Choose the best scenario on u(t) that minimizes M(s), including also the intervention-off scenario, and return to step 1, advancing the time s by 6 h.

The algorithm starts from s = 0 with ALAERT=OFF, ELAPSE=0, and u(0) = 0.

*Author contributions.* T.M. conceived the study and conducted the analyses with contributions from K.T., S.K., N.F., and A.O. All authors discussed and interpreted the results. The manuscript was written by all authors, with T.M. preparing the first draft.

Competing interests. The authors declare that they have no competing financial interests.

Acknowledgements. T.M. thanks Daiya Shiojiri for his support with the LETKF implementation. This study was partially supported by JST Moonshot R&D (JPMJMS2389) and JSPS KAKENHI Grant Number JP25K07942.

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
