# Peer review of "Bottom—up approach for mitigating extreme events with limited intervention options: a case study with Lorenz 96 model"

_EGUsphere, 2025_

## Author Comment (AC1)

**Reply to Referee #1 (Ms. Qin Huang)**

Takahito Mitsui[1], Shunji Kotsuki[2,3], Naoya Fujiwara[4], Astushi Okazaki[2,3], and Keita Tokuda[1]

[1]Faculty of Health Data Science, Juntendo Univerity, Urayasu, Japan
[2]Institute for Advanced Academic Research, Chiba University, Chiba, Japan
[3]Center for Environmental Remote Sensing, Chiba University, Chiba, Japan
[4]Graduate School of Information Sciences, Tohoku University, Sendai, Japan

**Correspondence:** Takahito Mitsui (takahito321@gmail.com)

Thank you very much for your detailed review of our manuscript and for providing valuable feedback. Below, we have listed your questions and comments (*italicized*), followed by our responses and proposed revisions to the manuscript (shown in violet). We believe that these changes improve the clarity and quality of our work.

*1. DA method comparison - This study uses LETKF for data assimilation, following Sun et al. (2023), which makes sense for a high-dimensional system like L96. Several other studies in this area (e.g., Miyoshi & Sun 2022; Kawasaki & Kotsuki 2024; Nagai et al. 2024) use EnKF, particularly with the lower-dimensional L63 model. It could be helpful to briefly clarify the rationale for choosing LETKF here, and short comments on whether the assimilation method affects control outcomes (even qualitatively) could be of interest to readers.*

We use the LETKF for data assimilation for the following reasons: (i) as you mentioned, the LETKF reduces computational costs for high-dimensional systems such as the Lorenz 96 model, mainly because it performs assimilation locally and operates in a low-dimensional ensemble space, avoiding expensive full-space computations and (ii) since we plan to use the SCALE-LETKF in future work, adopting the LETKF here provides a smoother transition. **In the revised version, we will include the first reason in the first paragraph of Section 2.2.**

*Control method comparison - While the manuscript positions the proposed method as a bottom-up alternative to top-down strategies like MPC, it does not include direct performance comparisons. A brief discussion of how the approach compares relative to recent MPC-based or other CSE studies could help contextualize its contributions. If direct comparisons are not feasible, outlining conceptual trade-offs or implementation differences would help clarify the novelty and practical significance.*

Thank you for pointing this out. We agree that our manuscript does not provide a direct performance comparison between our bottom-up strategy and top-down approaches such as Model Predictive Control (MPC). However, directly comparable MPC-based studies with the Lorenz 96 model have not been published yet. Therefore, we briefly outline the conceptual trade-offs and implementation differences here. Our bottom-up approach may be particularly suitable when intervention options are limited or discrete, while top-down methods like MPC are generally more effective when intervention options are continuous or nearly unlimited. In addition, our method is relatively simple to implement and offers interpretability, whereas MPC methods, though often more computationally intensive, can provide mathematically optimal control solutions under specified constraints. In the

revised paper, we will summarize these advantages and limitations in the Summary and Discussion section.

*Optimal control - The current method selecting intervention scenarios by minimizing the maximum ensemble outcome is*
30   *effective but not optimal. While other top-down strategies determine inputs using optimization minimizing costs, this utilizes limited intervention criteria.*

Yes, we agree that our bottom-up approach is not optimal. Top-down approaches typically determine optimal control inputs under cost constraints. Therefore, if sufficient computational resources and time are available, control-theoretic methods such as MPC would indeed be the most effective. Our study instead offers a complementary or backup approach for cases where
35   computational power is limited.

*Perturbation magnitude - In Fig. 4 (one-site) and Fig. 8 (two-site), the reported perturbation magnitudes (642.1 and 674.9, respectively) seem quite large. It would help to clarify the units or scale used, are these relative to system variability, or absolute values in state units?*

40   Thank you for this comment. Since the perturbation magnitude (i.e., intervention energy) is a time-integrated quantity, it is more useful to consider the intervention size. The intervention size, $u$, is itself dimensionless but can be compared with the other terms in Eq. (1). In fact, the intervention sizes, $u = 1.6$ for Fig. 4 as well as $u = 2.0$ for Fig. 8, are not particularly small when compared with the parameter $F = 8$ or the typical range of the state variable, $-12 \lesssim X_i \lesssim 16$ (Fig. 1a). On the other hand, a typical displacement caused by the intervention during the time step of $\Delta t = 0.01$ is $u \times \Delta t = 0.02$ for $u = 2.0$. This
45   is not particularly large when compared to the typical range $-12 \lesssim X_i \lesssim 16$. In Sun et al. (2023), the typical size of one-step displacement is $\alpha D_0 = 0.2 \times 0.1989 = 0.03978$ (see their Fig. 5). Therefore, the intervention size used in this study is comparable to, or smaller than, that used in the previous work. **We will mention these points in the revised manuscript**.

*Also, Fig. 4 references "operation energy" - what exactly does this refer to?*
50   We apologize for the confusion. The quantity "operation energy" refers to the intervention energy associated with changes in the intervention. However, it was mistakenly shown only in the figure and is no longer used in the present version of the manuscript. **We will delete it in the revised manuscript.**

*The average number of changes (22.4, 22.1) are shown as non-integers, since interventions are presumably discrete in time,*
55   *why are these fractional? Is this an average over ensembles or multiple trials?*

Yes exactly, the average number of changes is shown by a fractional because it is an average over ten 100-y segments.

*Overall, these suggestions are meant as optional additions - the manuscript is already very complete and well-structured. Including a bit more comparative context could further enhance clarity for readers unfamiliar with the broader control and*
60   *data assimilation literature.*

Thank you for your warm and helpful comments. We will improve the clarity of the paper in the revision.

**References**

Sun, Q., Miyoshi, T., and Richard, S.: Control simulation experiments of extreme events with the Lorenz-96 model, Nonlinear Process.
Geophys., 30, 117–128, https://doi.org/10.5194/npg-30-117-2023, 2023.

65

---

## Author Comment (AC2)

**Reply to Referee #2**

Takahito Mitsui[1], Shunji Kotsuki[2,3], Naoya Fujiwara[4], Astushi Okazaki[2,3], and Keita Tokuda[1]

[1]Faculty of Health Data Science, Juntendo Univerity, Urayasu, Japan
[2]Institute for Advanced Academic Research, Chiba University, Chiba, Japan
[3]Center for Environmental Remote Sensing, Chiba University, Chiba, Japan
[4]Graduate School of Information Sciences, Tohoku University, Sendai, Japan

**Correspondence:** Takahito Mitsui (takahito321@gmail.com)

Thank you very much for your detailed review of our manuscript and for providing valuable feedback. Below, we have listed your questions and comments (*italicized*), followed by our responses and proposed revisions to the manuscript (shown in violet). We believe that these changes improve the clarity and quality of our work.

5     *L115-120 The notation and explanation of the two intervention scenarios presented here are not so clear. In the "intervention-off" scenario, is there any intervention prior to time s?*

Thank you for pointing this out. For the "intervention-off" scenario planned in $s \leq t < s + 6$ h, there can be previously-planned interventions until $t = s + 6$ h, but any intervention is turned off from that point onward. **We will add this explanation just after Eq. (2), which defines the intervention-off scenario.**

10

    *In this scenario I assume u can be nonzero at different grid points. Is that correct?*

Yes, it is. More precisely, in the one-site intervention scenario, an intervention input $u_i(t)$ can take the non-zero value $u$ only at a single site, while in the two-site intervention scenario, $u_i(t)$ can be non-zero at two different sites, at most.

15     *In the one-site intervention scenario, u is nonzero at one grid point only but can also be nonzero before time s and after time s+6 hr. Is $u_i(t)$ constant for $t > s+6$ hr?*

Thank you for pointing this out. A simple answer is yes: the intervention input $u_i(t)$ is constant for $t \geq s + 6$ h in a scenario planned in each 6-h cycle. However, we can update the scenario in the following 6-h cycles. Therefore, the actual sequence of $u_i(t)$ can change in time for $t \geq s + 6$ h.

20

    *Is the intervention forcing u a constant (e.g., given that the forcing will be applied at grid point i, is the value of this forcing known a priori or is it something that will be optimized)?*

Yes, the intervention size, $u$, is a predetermined constant parameter here. The optimization of $u$ is omitted in the present study to reduce computational cost. **For clarity, we will add the following sentence after Eq. (3): "where $u$ is a constant**
25 **parameter representing the actual intervention size."** This assumption may be justified by the following consideration: since human influence is small compared to the dynamics of weather systems, it is reasonable to assume that the intervention

operates at its maximal feasible strength.

*Figure 2: What do the second panels of Figure a) i) and ii) mean? Also in Figure b)iii), ALERT instead of ALEAT.*

The first and the second panels in Fig. 2a (i) and (ii) show two different cases. In the second panels of Fig. 2a (i) and (ii), there is no intervention over $s \leq t < s + 6$ h since it was not selected in the previous 6-h cycle. On the other hand, the first panels show the cases where there is an intervention over $s \leq t < s + 6$ h. **We will add the following sentence in the caption: "The lower panels in both (a-i) and (a–ii) show the cases where no intervention over $t \geq s$ was selected in the previous 6-hour cycle."** Also the typo is corrected. Thank you.

*L137 Multi-scenario ensemble forecast and local intervention:*

Thank you: **We have changed from 'milti' to 'Multi'.**

*L145 In the sentence starting with "Other criteria . . . " I can not clearly follow the difference from the previous criterion.*

Sorry for this confusion. **We will replace 'the expected maximum' by 'the ensemble-mean of the maximum value across all sites'.**

*L152 In this part the approach to evaluate sampling errors in the resulting scores is discussed. Why did the authors decide to work with smaller samples instead of applying bootstrap to the whole sample?*

We worked with the ten 100-y samples as in Sun et al. (2023). This allows us a clear comparison with that work. **However, in the revised manuscript, we will mention the dependence of the results on sampling methods.**

*L167 "This metric is particularly relevant . . . " My impression is the opposite: when the intervention is static, scenarios can not be changed, and this metric is not relevant.*

Thank you for pointing this out. Indeed, if the intervention is static (i.e., time-invariant), the scenarios cannot be changed. We believe that this apparent contradiction is solved if 'static' is removed form the corresponding sentence.

*In the discussion of Figure 4, the method of Sun et al. is compared with the new method. In the first comparison the success rate is much higher for the new method. However, this is done for a larger intervention size and for a shorter forecast window than in Sun et al. Figures 5 and 6 show that under similar intervention energy and forecast length, Sun et al.'s method are closer to the results obtained with the proposed method. A similar comparison is presented in the abstract and in the conclusions; however, it is unclear if the numbers commented on in the abstract correspond to the numbers in this section. If so, the claim of the abstract and the conclusions does not seem to be a clear comparison with Sun et al.'s approach.*

Thank you for this comment. As you suggested, our comparison with Sun et al. (2023) was not entirely fair. **In the revised manuscript, we will modify the concluding statement to focus on our own achievement rather than making potentially debatable comparisons**:

(Before) **The success rate of our method is markedly higher than that of Sun et al.'s method,** reaching 94% even when applying interventions at one site per step, ...

(After) **Our method achieves a high success rate,** reaching 94% even when applying interventions at one site per step, ...

65     We also note your observation that our success rate is comparable with that of Sun et al. (2023) under similar intervention energy.

    *L190: "... necessary is not guaranteed". Can this be assessed from the previous experiment? The distribution of the distance of the optimal interventions with respect to the location of the extreme event can be obtained and analyzed to support this*
70 *claim.*

    The original statement, "Whether interventions across all 40 sites are feasible or (if feasible) necessary is not guaranteed," was potentially misleading. What we actually intended was that examining all possible intervention combinations is infeasible in real-world operations. **We will revise the statement as follows: "Assuming real-world applications, interventions across all sites may be neither feasible nor necessary."**

75

    *Figure 7. Panel c describes the number of scenario changes. This metric seems to grow rapidly from 1 intervention-eligible site to 3. However, I wonder what the behavior would be if the distance associated with each change is also taken into account. It would make sense to distinguish between many small changes and few larger changes (also considering that sometimes the change needs to be done in a small time frame).*

80     We assume that the reviewer is concerned with spatial distances between consecutive intervention sites. These distances may increase as the number of intervention-eligible sites increases. In the present study, this factor is not included in the cost estimation. **In the revised manuscript, we will mention that increasing the number of intervention-eligible sites may result in higher transportation costs.**

85     *L211 complete instead of complate.*

    Thank you. We correct it.

    *L214 Figure 11?*

    Yes, we change it from Fig. 10 to Fig. 11. Thank you.

90

    *L215: Why is the ensemble size increased in this experiment? I understand that the localization scale has to be adjusted when the observation network is changed; however, increasing the ensemble size is assumed to always lead to a better performance of the filter (particularly at these relatively small ensemble sizes), but always limited by the available computational power.*

    We increased the number of ensemble size only a little from 10 to 11 in order to have a better performance. However, as you
95 point out, it makes the interpretation of the results less clear. **Thus, in the revised manuscript, we will show also the cases with 10 ensemble members to clearly show the effect of partial observation.**

**References**

Sun, Q., Miyoshi, T., and Richard, S.: Control simulation experiments of extreme events with the Lorenz-96 model, Nonlinear Process. Geophys., 30, 117–128, https://doi.org/10.5194/npg-30-117-2023, 2023.

---

## Author Response (AR1)

Dr. Takahito Mitsui Faculty of Health Data Science Juntendo University, Japan takahito321@gmail.com

14 June 2025

Dear Prof. Natale Alberto Carrassi,

Thank you very much for reviewing our manuscript entitled "Bottom-up approach for mitigating extreme events with limited intervention options: a case study with the Lorenz 96 model." We are pleased to resubmit a revised version of the manuscript, which has been updated in accordance with the referees' comments.

Below, we provide a summary of the main changes, followed by our detailed, point-by-point responses to the referees' comments. We believe that these revisions have significantly improved the clarity and overall quality of the manuscript.

**Summary of the main changes**

- The title "... under limited options: a case study with Lorenz 96" is slightly modified as "... with limited options: a case study with Lorenz 96 model".
- Following a comment by Reviewer #2, Figure 11 is updated to include both the results for the standard LETKF parameters in the present study and those for the optimized parameters. Please see the response to the last comment below.
- As Referee #2 pointed out, our comparison of success rates with Sun et al. (2023) was not entirely fair, as the results were obtained under different intervention energies. In the revised manuscript, we have modified the concluding statement to emphasize our own achievements rather than making potentially debatable comparisons (see p. 5 of this document).

In the following, the referees' comments are shown in *italics*, our responses are presented in violet, and the corresponding revisions made to the manuscript are indicated in **bold**.

**Reply to Referee #1**

1. DA method comparison - This study uses LETKF for data assimilation, following Sun et al. (2023), which makes sense for a high-dimensional system like L96. Several other studies in this area (e.g., Miyoshi & Sun 2022; Kawasaki & Kotsuki 2024; Nagai et al. 2024) use EnKF, particularly with the lower-dimensional L63 model. It could be helpful to briefly clarify the rationale for choosing LETKF here, and short comments on whether the assimilation method affects control outcomes (even qualitatively) could be of interest to readers.

We appreciate this reviewer's comment. In the revised manuscript, we have mentioned the rationale for using the LETKF and also whether the assimilation method affects control outcomes (Section 2.2, Lines 76–79 and 103–107):

"As in Sun et al. (2023), we adopt the Local Ensemble Transform Kalman Filter (LETKF) (Hunt et al., 2007), a variant of the Ensemble Kalman Filter, for data assimilation due to its efficiency and scalability in high-dimensional settings such as numerical weather prediction models (Lien et al., 2017). [...] In sum, the LETKF

performs data assimilation locally and operates within a low-dimensional ensemble space, thereby avoiding expensive full-space computations and reducing the computational cost in high-dimensional systems. Although we use the LETKF in this study, comparable control performance may be achieved with other data assimilation methods, such as the Serial Ensemble Square Root Filter, provided that state estimation is performed as accurately as with the LETKF (Miyoshi, 2005)."

Control method comparison - While the manuscript positions the proposed method as a bottomup alternative to top-down strategies like MPC, it does not include direct performance comparisons. A brief discussion of how the approach compares relative to recent MPC-based or other CSE studies could help contextualize its contributions. If direct comparisons are not feasible, outlining conceptual trade-offs or implementation differences would help clarify the novelty and practical significance.

We agree that our manuscript does not provide a direct performance comparison between our bottom-up strategy and top-down approaches such as Model Predictive Control (MPC). However, directly comparable MPC-based studies with the Lorenz 96 model have not been published yet. Therefore, we briefly outline the conceptual trade-offs and implementation differences in Section 4, Lines 265–268: "Our bottom-up approach may be particularly well-suited when intervention options can be limited or discretized. In contrast, top-down methods such as MPC are generally more effective when intervention options are continuous or nearly unlimited. Moreover, our method is relatively simple to implement and highly interpretable, whereas methods like MPC, although typically more computationally demanding, can provide mathematically optimal control solutions under given constraints."

Optimal control - The current method selecting intervention scenarios by minimizing the maximum ensemble outcome is effective but not optimal. While other top-down strategies determine inputs using optimization minimizing costs, this utilizes limited intervention criteria.

Yes, we agree that our bottom-up approach is not optimal. Top-down approaches typically determine optimal control inputs under cost constraints. Therefore, if sufficient computational resources and time are available, control-theoretic methods such as MPC would indeed be the most effective. Our study instead offers a complementary approach for cases where computational power is limited. The response is already included in the above one.

Perturbation magnitude - In Fig. 4 (one-site) and Fig. 8 (two-site), the reported perturbation magnitudes (642.1 and 674.9, respectively) seem quite large. It would help to clarify the units or scale used, are these relative to system variability, or absolute values in state units?

To gain intuition about the perturbation magnitude, one can compare the intervention size, u, with the other terms in Eq. (1). Actually, the intervention size u = 1.6 that achieves successful control (Fig. 4) is not particularly small relative to the parameter F = 8 or the typical state range  $-12 \lesssim X_i \lesssim 16$ . This corresponds to an intervention-induced displacement of one step, with  $u\Delta t = 0.016$  and  $\Delta t = 0.01$ . In comparison, Sun et al. (2023) employed intervention-induced displacements  $\alpha D_0$ , ranging from 0.01989 to 0.1989. Thus, the intervention magnitude in our study is comparable to or smaller than that in Sun et al. (2023). We have added this explanation in lines 198–202 in Section 3.1.

Also, Fig. 4 references "operation energy" - what exactly does this refer to?

We apologize for the confusion. The quantity "operation energy" is the same as "intervention energy". It was mistakenly shown in the figure and is no longer used in the present version of the

manuscript. We have deleted it in the revised manuscript.

The average number of changes (22.4, 22.1) are shown as non-integers, since interventions are presumably discrete in time, why are these fractional? Is this an average over ensembles or multiple trials?

Yes, exactly. The average number of changes is shown as a fractional value because it is averaged over ten 100-year segments. To be clear, we have added the following sentence in the caption of Fig. 5: "Points represent the average values, and error bars indicate one standard deviation across 10 experimental results with different initial conditions."

Overall, these suggestions are meant as optional additions - the manuscript is already very complete and well-structured. Including a bit more comparative context could further enhance clarity for readers unfamiliar with the broader control and data assimilation literature.

We believe that the present revisions significantly improve the clarity.

**Reply to Referee #2**

L115-120 The notation and explanation of the two intervention scenarios presented here are not so clear. In the "intervention-off" scenario, is there any intervention prior to time s?

Thank you for pointing this out. For the "intervention-off" scenario planned in  $s \le t < s + 6$  h, there can be previously-planned interventions until t = s + 6 h, but any intervention is turned off from that point onward. We have added this explanation just after Eq. (2), which defines the intervention-off scenario as follows: "Here  $u_i(s)$  can be either zero or non-zero depending on the preceding scenario chosen during s-6 h  $\le t < s$ , but any intervention is switched off from t=s+6 h."

In this scenario I assume u can be nonzero at different grid points. Is that correct?

Yes, it is. More precisely, in the one-site intervention scenario, an intervention input  $u_i(t)$  can take the non-zero value u only at a single site, while in the two-site intervention scenario,  $u_i(t)$  can be non-zero at two different sites, at most.

In the one-site intervention scenario, u is nonzero at one grid point only but can also be nonzero before time s and after time s+6 hr. Is  $u_i(t)$  constant for t>s+6 hr?

A simple answer is yes: the intervention input  $u_i(t)$  is constant for  $t \ge s + 6$  h in a scenario planned in each 6-h cycle. However, we can update the scenario in the following 6-h cycles. Therefore, the actual sequence of  $u_i(t)$  can change in time for  $t \ge s + 6$  h.

Is the intervention forcing u a constant (e.g., given that the forcing will be applied at grid point i, is the value of this forcing known a priori or is it something that will be optimized)?

Yes, the intervention size, u, is a predetermined constant parameter here. The optimization of u is omitted in the present study to reduce computational cost. For clarity, we have added the following sentence after Eq. (3): "where u is a constant parameter representing the actual intervention size." This assumption may be justified by the following consideration: since human influence is small compared to the dynamics of weather systems, it is reasonable to assume that the intervention operates at its maximal feasible strength.

Figure 2: What do the second panels of Figure a) i) and ii) mean? Also in Figure b)iii), ALERT instead of ALEAT.

The first and the second panels in Fig. 2a (i) and (ii) show two different cases. In the second panels of Fig. 2a (i) and (ii), there is no intervention over  $s \le t

Figure 1: Performance metrics versus input size u of the proposed control method at a single intervention site: (a) success rate, (b) intervention energy, and (c) number of scenario changes. The prediction horizon T is 7 d. Error bars are computed using two different approaches. The first is the original approach, which uses ten 100-year samples from the 1000-year data series to compute the standard deviation of the metrics ('stats 1'). The second uses one hundred 100-year samples obtained from the same 1000-year series by sliding the sampling window every 10 years ('stats 2').

L167 "This metric is particularly relevant ..." My impression is the opposite: when the intervention is static, scenarios can not be changed, and this metric is not relevant.

Indeed, if the intervention is static (i.e., time-invariant), the scenarios cannot be changed. In order to solve this apparent contradiction, we have removed 'static' form the corresponding sentence. We believe that this solves the issue.

In the discussion of Figure 4, the method of Sun et al. is compared with the new method. In the first comparison the success rate is much higher for the new method. However, this is done for a larger intervention size and for a shorter forecast window than in Sun et al. Figures 5 and 6 show that under similar intervention energy and forecast length, Sun et al.'s method are closer to the results obtained with the proposed method. A similar comparison is presented in the abstract and in the conclusions; however, it is unclear if the numbers commented on in the abstract correspond to the numbers in this section. If so, the claim of the abstract and the conclusions does not seem to be a clear comparison with Sun et al.'s approach.

As Referee #2 pointed out, our comparison of success rates with Sun et al. (2023) was not entirely fair, as the results were obtained under different intervention energies. In the revised manuscript, we have modified the concluding statement to focus on our own achievement rather than making potentially debatable comparisons.

In the abstract:

(Before) The success rate of our method is markedly higher than that of Sun et al.'s method ... (After) Our method achieves high success rates ...

Also for a further fair comparison, we have mentioned the observation by the referee: "With the same intervention energy (in the case u = 0.5), our success rate ( $\sim$ 45%) is lower than 60% reported in Sun et al. (2023)" (lines 194–195 in Section 3.1 and lines 248–250 in the summary).

L190: "... necessary is not quaranteed". Can this be assessed from the previous experiment?

The distribution of the distance of the optimal interventions with respect to the location of the extreme event can be obtained and analyzed to support this claim.

The original statement, "Whether interventions across all 40 sites are feasible or (if feasible) necessary is not guaranteed," was potentially misleading. What we actually intended was that examining all possible intervention combinations is infeasible in real-world operations. We have revised the statement at line 207: "Assuming real-world applications, interventions across all sites could be neither feasible nor necessary."

Figure 7. Panel c describes the number of scenario changes. This metric seems to grow rapidly from 1 intervention-eligible site to 3. However, I wonder what the behavior would be if the distance associated with each change is also taken into account. It would make sense to distinguish between many small changes and few larger changes (also considering that sometimes the change needs to be done in a small time frame).

We assume that the reviewer is concerned with spatial distances between consecutive intervention sites. These distances may increase as the number of intervention-eligible sites increases. In the present study, this factor is not included in the cost estimation. In the revised manuscript, we have noted as follows (lines 176-177). "Obviously, other types of cost functions can be considered—for example, changing the intervention site to a more distant location may incur higher costs. However, in this study, we focus on the three performance metrics mentioned above."

L211 complete instead of complete. Corrected.

L214 Figure 11?

We have corrected it from Fig. 10 to Fig. 11.

L215: Why is the ensemble size increased in this experiment? I understand that the localization scale has to be adjusted when the observation network is changed; however, increasing the ensemble size is assumed to always lead to a better performance of the filter (particularly at these relatively small ensemble sizes), but always limited by the available computational power.

We increased the number of ensemble size only a little from 10 to 11 in order to have a better performance. However, as the referee points out, it makes the interpretation of the results less clear. Thus, in the revised manuscript, we have shown the cases with 10 ensemble members to clearly show the effect of partial observation in new Fig. 11. Accordingly, we have modified text in the last paragraph of Section 3.4 (Lines 230–237).

**References**

Brian R Hunt, Eric J Kostelich, and Istvan Szunyogh. Efficient data assimilation for spatiotemporal chaos: A local ensemble transform kalman filter. *Phys. D: Nonlinear Phenom.*, 230(1-2):112–126, 2007. doi: 10.1016/j.physd.2006.11.008.

Guo-Yuan Lien, Takemasa Miyoshi, Seiya Nishizawa, Ryuji Yoshida, Hisashi Yashiro, Sachiho A Adachi, Tsuyoshi Yamaura, and Hirofumi Tomita. The near-real-time scale-letkf system: A case of the september 2015 kanto-tohoku heavy rainfall. *Sola*, 13:1–6, 2017.

- Takemasa Miyoshi. Ensemble Kalman filter experiments with a primitive-equation global model. University of Maryland, College Park, 2005.
- Qiwen Sun, Takemasa Miyoshi, and Serge Richard. Control simulation experiments of extreme events with the lorenz-96 model. Nonlinear Process. Geophys., 30:117-128, 2023. doi: 10.5194/npg-30-117-2023.